# Matrix Regeneration Ability In Situ Induced by a Silk Fibroin Small-Caliber Artificial Blood Vessel In Vivo

**DOI:** 10.3390/polym14183754

**Published:** 2022-09-08

**Authors:** Helei Li, Mengnan Dai, Meng Li, Lingpeng Meng, Yangxiao Yu, Jianmei Xu, Fenglin Dong, Qingmin Fan, Yin Yin, Aiqing Wang, Jiannan Wang

**Affiliations:** 1College of Textile and Clothing Engineering, Soochow University, Suzhou 215123, China; 2Department of Ultrasonography, First Affiliated Hospital, Soochow University, Suzhou 215006, China; 3Department of Medicine, Soochow University, Suzhou 215123, China

**Keywords:** silk fibroin, small-caliber artificial vessels, ECM, matrix fiber self-assembly

## Abstract

The success of a small-caliber artificial vascular graft in the host in order to obtain functional tissue regeneration and remodeling remains a great challenge in clinical application. In our previous work, a silk-based, small-caliber tubular scaffold (SFTS) showed excellent mechanical properties, long-term patency and rapid endothelialization capabilities. On this basis, the aim of the present study was to evaluate the vascular reconstruction process after implantation to replace the common carotid artery in rabbits. The new tissue on both sides of the SFTSs at 1 month was clearly observed. Inside the SFTSs, the extracellular matrix (ECM) was deposited on the pore wall at 1 month and continued to increase during the follow-up period. The self-assembled collagen fibers and elastic fibers were clearly visible in a circumferential arrangement at 6 months and were similar to autologous blood vessels. The positive expression rate of Lysyl oxidase-1 (LOXL-1) was positively correlated with the formation and maturity of collagen fibers and elastic fibers. In summary, the findings of the tissue regeneration processes indicated that the bionic SFTSs induced in situ angiogenesis in defects.

## 1. Introduction

Vascular diseases (such as hypertension and myocardial infarction) consistently threaten human life, the global disease burden and subsequent deaths are projected to escalate exponentially, with an estimated 23.4 million deaths in 2030 [1,2]. Replacing diseased or narrowed blood vessels is the most effective way to treat these conditions. There is a significant need for small-caliber vascular grafts. Autologous blood vessels, such as the saphenous vein, internal thoracic artery, and radial artery, are considered optimal for vascular transplantation [3]. However, the use of these vessels is limited by the lack of available sources and morbidity of the donor sites [4,5]. Large-caliber (>6 mm) vascular prostheses composed of synthetic materials (e.g., polyester (PET) and expanded polytetrafluoroethylene (ePTFE)) have long been used in vascular surgery; nevertheless, due to the inherent hydrophobic property of an extremely large surface area, these materials can not only substantially increase the adsorption of platelets and plasma proteins but also decrease endothelial cells’ (ECs) adhesion and growth on their surface [6,7]. Therefore, PET and ePTFE are not suitable for application in small-caliber (<6 mm), artificial blood vessels, even though they have been used clinically for a long time [8]. To overcome these limitations, there is a trend in using natural biomaterials for the development of artificial blood vessels.

Natural biomaterials are mainly derived from animals and plants and are degradable biomacromolecules. Therefore, natural animal proteins (collagens/gelatin, elastin, fibrinogen, and silk fibroin) are conducive to cell adhesion, migration, and proliferation [9,10,11]. Collagen/gelatin, elastin, and fibrinogen are the major components of ECM, which make them the primary choice for the construction of artificial blood vessels [12]. They are mainly used in nanofiber electrospinning technology to construct tubular scaffolds [13,14]. However, insufficient mechanical properties do not support their function in vivo [15]. Studies have also been performed to twist the collagen and elastin fibers to obtain filaments, and then fabricate single-layer knitted grafts in order to improve the mechanical properties. Unfortunately, their performance in vivo has not been further published [14]. Another method used is to combine these with synthetic polymer materials to compensate for the mechanical deficiency of natural polymer materials [16,17,18]. The mechanical properties of these grafts were preserved for one month after replacement of the rabbit common carotid artery [17], however, the grafts did not induce cell growth [18].

Silk fibroin (SF) is a natural high-molecular-weight polymer and is considered an ideal material for tissue-engineering scaffolds [19,20]. Compared with collagen/gelatin, elastin, and fibrinogen, silk is rich in sources and composed of approximately 25% sericin and 75% SF fiber, which possesses excellent mechanical properties (breaking strength (300–740 MPa) and breaking elongation (7–26%)) [21,22]. The silk fibers are easily prepared into regenerated SF with low immunogenicity, anticoagulation, and biodegradability [23,24]. SF materials not only possess the same biological activity as collagen and elastin but also exhibit a similar electronegativity to the inner surface of natural blood vessels compared with PET and ePTFE. SF can be degraded by matrix metalloproteinase, collagenase, elastase, and so on in vivo. Regenerated SF materials degrade rapidly, but high crystalline silk fibers degrade slowly.

In recent years, many scholars have carried out studies on SF materials. Small-caliber blood vessels constructed via electrospinning technology can support cell adhesion in vitro [25], but there was little cell infiltration when these vessels were inserted into the back of rats [2]. It is well-known that natural blood vessels are subjected to the shearing force of blood in the body [26], and the results of vessel implantation in the back of rats have no clinical significance. A double-layer, small-caliber artificial blood vessel prepared by the combination of electrospinning and freeze-drying was used to replace the abdominal aorta in rats for 8 weeks, and histological analysis revealed neo-tissue formation, host cell infiltration, and graft remodeling in terms of ECM turnover [27]. Natural SF fibers were also incorporated into vascular tissue engineering. A small-caliber, artificial vascular scaffold prepared by SF fibers was implanted in male Sprague–Dawley (SD) rats, resulting in long-term patency for one year and tissue formation [28,29].

Following tissue-engineering blood vessel transplantation, the vessel must be able to maintain temporary mechanical properties, and then gradually be replaced by cell infiltration and tissue regeneration. Finally, new blood vessels are reconstructed in situ to achieve function. A vascular graft requires a suitable three-dimensional (3D) microenvironment for cell migration, colonization, and growth [30]. Using electrospinning technology, it is difficult to form a 3D porous scaffold, which is not conducive to cell survival [31], and the vessel scaffolds also lacked mechanical properties. Freeze drying and 3D printing are effective methods for providing the scaffold with a 3D porous structure [32,33]. Braiding, a traditional textile technique, can be considered to improve mechanical properties in the construction of artificial blood vessels. Therefore, a silk-based, small-caliber tubular scaffold (SFTS) constructed by combining freeze-drying technology and braiding was developed in our previous research [34,35,36,37], which possessed excellent mechanical properties, long-term patency (for 1.5 years), and achieved rapid endothelialization and smooth muscle cells’ (SMCs) infiltration and proliferation after transplantation to replace the rabbit common carotid artery [38,39].

The long-term stability of artificial blood vessels is a vital indicator after transplantation. To the best of our knowledge, SF grafts with an inlaid braiding were implanted in SD rats for 12 and 18 months as reported by Nakazawa et al. [28] and Enomoto et al. [40], while the dynamic process of tissue regeneration has not been studied in detail. The present study mainly focused on the tissue regeneration process (1 W, …1 M, …12 M) within 12 months after surgery in rabbits, including the synthesis and distribution of ECM and the occurrence and maturation of collagen fibers and elastic fibers. It is noteworthy that these studies will be crucial for the breakthrough of small-caliber, tissue-engineered blood vessels in clinical application, the realization of patient rehabilitation, and the improvement of tissue-engineered blood vessel construction methods.

## 2. Materials and Methods

### 2.1. Preparation of SFTSs

SFTSs were prepared as described previously [38]. In brief, raw silk was degummed with 0.06% of sodium carbonate (0.06%), dissolved in 9.3 mol/L lithium bromide, then dialyzed against distilled water at 4 °C for 3 days, and concentrated and filtered to obtain SF aqueous solution (80 mg/mL). A silk tubular fabric was braided with a 3 mm inner diameter by twisting 24 shares of degummed 3 × 2 silk yarns. The 3 × 2 silk yarn consisted of two-ply yarns composed of two strands of thrown silk, each with three strands of 20/22 D raw silk. Then, the tubular fabrics were immersed in a mold containing SF aqueous solution, and then freeze-dried to form SFTSs. Following sterilization by 25 kGy of Co^60^ gamma irradiation, the SFTSs were stored at 4 °C.

### 2.2. In Vivo Implantation of SFTSs

Animal experiments were performed by replacing the common carotid artery of healthy male New Zealand white rabbits (2.5 kg, 10 weeks old). All procedures were approved by the animal ethics committee of Soochow University (No. ECSU-2019000135). As described in our previous work [38], the rabbits were anesthetized with 30% pentobarbital sodium via intravenous injection, and the right common carotid artery was excised and replaced with an SFTS (15 mm in length and 3 mm in inner diameter) using an end-to-end anastomosis. The rabbits were followed-up for 12 months. The SFTSs were explanted at 1 week (1 W), 2 weeks (2 W), 1 month (1 M), 3 months (3 M), 6 months (6 M), 9 months (9 M), and 12 months (12 M) (n = 6 per time point), respectively.

### 2.3. Scanning Electron Microscopy

The SFTSs taken from the rabbit common carotid artery were fixed with 4% paraformaldehyde for 30 min and soaked in 0.1 M phosphate buffer saline for 15 min (repeated 3 times alternately), and then dehydrated with gradient ethanol (50%, 70%, 80%, 90%, 95% and 100%) for 15 min, respectively. Finally, the specimen was air-dried and pasted on the workbench, and the morphological structure was observed using scanning electron microscopy (SEM, S-4800, Hitachi, Tokyo, Japan) after being sputter-coated with gold.

### 2.4. Histological Analysis

SFTSs were removed, fixed with 4% paraformaldehyde, and embedded in paraffin. Five μm sections were cut, deparaffinized, rehydrated, and then washed briefly with distilled water. The sections were stained with hematoxylin-eosin (HE) (Sinopharm, Beijing, China), Massons, and WVG test kits (Yuanye Biotech, Shanghai, China) to observe cells, collagen fibers, and elastic fibers using an IX51 inverted microscope (Olympus, Tokyo, Japan).

### 2.5. Immunohistochemical Staining

The histologic sections were prepared using the same method as that for histological analysis. Then, sections were incubated for 16 h at 4 °C with rabbit anti-human elastin, type III collagen antibody (Bioss, Beijing, China), rabbit anti-human LOXL-1 antibody (Novusbio, Littleton, CO, USA) (corresponding donkey anti-rabbit IgG labeled with horseradish peroxidase secondary antibody; Bioss), or mouse anti-human type I collagen antibody (GeneTex, San Antonio, TE, USA) (corresponding rabbit anti-mouse IgG labeled with horseradish peroxidase secondary antibody; Bioss), respectively. Table 1 shows the dilution factor of corresponding antibodies. After overnight incubation and washing, the secondary antibodies were added and incubated for 1 h at 37 °C. Finally, all sections were counterstained with hematoxylin and observed using the IX51 inverted microscope (Olympus). Quantification was performed using Image J software (Version 1.8.0, US National Institute of Health, Bethesda, MA, USA, http://rsb.info.nih.gov/ij/ (accessed on 21 November 2018). Ten images for each time point were used to calculate the positive expression rate of matrix proteins and cytokines, in which the percentage of the brown positive expression area occupied the entire interface.

### 2.6. Real-Time Quantitative PCR (RT-PCR) Analysis

The SFTS samples were quickly frozen in liquid nitrogen and ground, and then total RNA extraction and reverse transcription were performed according to the standard method of the centrifugal column total RNA purification kit (Sangon Biotech, Shanghai, China) and the M-MuLV first-strand cDNA synthesis kit (M-MuLV, Sangon Biotech), respectively. The content of RNA was determined using an ultraviolet spectrophotometer (Bio-Rad, Hercules, CA, USA). Primers for each protein or cytokine were synthesized by Sangon Biotech Co., Ltd. (Shanghai, China), and the amplification efficiency of each pair of primers was measured as shown in Table 2. Quantitative real-time PCR was carried out using a Step One Plus Real-Time PCR System (Applied Biosystems, Foster City, CA, USA), and the cycling conditions were: pre-denaturation at 95 °C for 15 min, followed by 40 cycles of denaturation at 95 °C for 30 s, annealing at the appropriate temperature (52–60 °C, Table 2) for 30 s, and elongation at 72 °C for 30 s. The housekeeping gene GAPDH was used as the internal reference gene. Finally, the autologous blood vessel was used as a control sample for data processing (the relative quantitative calculation was based on the autologous blood vessel percentage). The 2^−^^ΔΔCT^ method was used for quantitative analysis.

### 2.7. Statistical Analysis

All the quantitative results are presented in mean ± standard deviation (SD). Statistical significance was determined by one-way analysis of variance using SPSS software (Version 20, IBM, Chicago, IL, USA). Significance was set at * *p* < 0.05 and ** *p* < 0.01.

## 3. Results and Discussion

### 3.1. Morphology Observation by SEM and HE

SFTSs are considered to temporarily replace the ECM to support cell infiltration and growth. With the synthesis and deposition of ECM and the gradual degradation of SFTSs, tissue regeneration is finally achieved. From the SEM results, the three-layer structure (intima, media, and adventitia) of the animal natural blood vessel wall can be clearly seen (Figure 1A-a,B-a). At 1 M after implantation, the internal porous structure of implanted SFTSs (pink arrows) was visible, although it was significantly deformed compared with the non-implanted SFTS. Nevertheless, a thin layer of regenerated tissue was formed on the outside (Figure 1A-b). A magnified view shows the deposition of ECM on the pore wall in SFTSs (black arrows), while the pore walls of non-implanted SFTS were smooth. Compared with that at 1 M, the ECM filled the entire SFTS at 3 M after implantation, and the porous structure inside the SFTSs almost completely disappeared, but the regenerated tissue structure appeared to be loose, especially in the interior (Figure 1(A-c)). At 6 M after implantation, the entire SFTS (regenerated tissue) wall was tight, and the ECM was arranged in an orderly manner (Figure 1(A-d)). With the extension of implantation time, the regenerated tissue matured further and formed regular matrix slices similar to the morphology of autologous blood vessels (Figure 1(A-e)). HE staining showed similar results (Figure 1B). Cell infiltration is a prerequisite for tissue regeneration. From 1 M after implantation, vascular cells (blue cell nuclei) were uniformly distributed in the pores of SFTS, and new tissues formed in a sandwich from both sides of the SFTS (Figure 1(B-b, B-c)). At 6 M after implantation, the new tissues with regular morphological structure replaced the porous SFTS, and the cells were orderly distributed in the ECM along the circumferential direction (Figure 1(B-d)). The regenerated tissue structure in the circumferential direction was further self-assembled and matured and looked like autologous vascular tissue at 9 M (Figure 1(B-e)).

### 3.2. Deposition and Distribution of ECM after SFTSs Implantation

#### 3.2.1. Elastin

The ECM of autologous blood vessels is mainly composed of collagen and elastin [41]. Elastin is the precursor protein of elastic fibers, which bears the strain capacity of vasoconstriction and vasodilation [42]. Thus, it is important to study the synthesis, deposition, and distribution of elastin during the dynamic process of tissue regeneration. Figure 2A shows that elastin in the autogenous vessel was evenly distributed from the intima to the adventitia. The SFTS was also non-specifically stained brown (Figure 2B). At the initial stage of implantation, non-specific staining derived from many inflammatory cells and their secreted factors were clearly visible in the SFTS due to the stimulation of external materials at the wound. Subsequently, SMCs infiltrated the porous SFTS at 2 W (Figure 2D). At 1 M after implantation, the morphological structure of the SFTS looked similar to that of the non-implanted SFTS, which remained a clear porous structure. A large number of SMCs were distributed in the SFTS, and simultaneously, elastin was secreted and deposited on the pore wall. However, at this time, the generated matrix protein was scattered and disordered. Although the porous SF did not degrade significantly, the material was fragile and fell off in the cavity when made into slices (Figure 2E, followed by Figure 3E and Figure 4E). In the first 3 M after implantation, the positive expression of elastin increased significantly but not regularly. From 6 M to 12 M, the brown-stained elastin was evenly distributed in the tube wall and in an ordered arrangement along the circumference of the SFTSs (regenerative tissue) (Figure 2G–I). The porous SF material was completely degraded at 6 M after transplantation compared with that at 3 M. The results of the semi-quantitative analysis according to immunohistochemical staining showed that the positive expression of elastin continued to increase 9 M after implantation and entered a state of dynamic equilibrium (Figure 2J).

#### 3.2.2. Collagen

Compared with elastin, collagen is relatively abundant in arterial blood vessels. Collagen fibers in vascular tissue have tensile strength, and mainly include fibrous collagen type I (COL-I) and type III (COL-III), which account for 60% and 30% of the total collagen in blood vessels, respectively [43]. The COL-I in autologous blood vessels was mainly distributed in the media and adventitia (Figure 3A). There was also a wide distribution inside the SFTSs at the initial stage of implantation (1 W and 2 W), which showed non-specific staining (Figure 3C,D). After 1 M, uniform COL-I deposition on the pore wall was observed (Figure 3E). At 3 M after implantation, it can be seen that a partial area of positive expression COL-I was ordered (Figure 3F). At 6 M after implantation, COL-I was evenly distributed along the circumferential direction (Figure 3G), and its arrangement became more ordered and regular (Figure 3H–I). The quantitative immunohistochemistry results show that the positive expression rate of COL-I was higher than that of elastin. With the extension of implantation time, the positive expression rate of COL-I continued to increase as well as that of elastin, reaching a stable state at 9 M after implantation (Figure 3J).

COL-III is mainly found in the intima and adventitia of the blood vessel wall, which provides tensile strength for the blood vessel and plays a role in regulating the self-assembly of COL-I fibers [44,45]. COL-III also plays a part in cell adhesion, migration, proliferation, and differentiation through its interaction with integrins [46]. A brown area was seen in the wall of SFTSs at 1 M after implantation (Figure 4E). This brown area deepened, and the distribution area increased at 3 M (Figure 4F) and became ordered along the circumferential layers at 6 M (Figure 4G), which was consistent with the deposition and distribution of elastin and COL-I, but the amount of deposition was relatively low. The synthesis of COL-III increased significantly and showed a dark brown positive expression at 12 M (Figure 4I). Semi-quantitative results show that the deposited quantity of COL-III in the SFTS gradually increased, reaching a stable state at 9 M (Figure 4J).

### 3.3. The Expression Level of ECM by RT-PCR

The mRNA expression levels of elastin, COL-I, and COL-III in infiltrated cells in the SFTSs were determined by RT-PCR, as shown in Figure 5. Due to autologous blood vessel trauma and contact with foreign materials, SMCs were stimulated to proliferate and secrete ECM at the suture site and were in a secretory phenotype [39]. At the same time, stem cells (such as adipose stem cells [47] and bone marrow mesenchymal stem cells [48]) on both sides migrated into the SFTSs and differentiated into vascular cells, which exhibited high proliferation and synthesis of ECM [49]. So, the mRNA expression levels of elastin (1 W) and COL-I (1 M) were up-regulated at the early stage of implantation and then began to decrease but rebounded at 6 M, which was possibly related to the rapid degradation of porous SF at 3 M after implantation (Figure 5A). At 3 M, the pore structure was significantly larger, which was caused by the fracture of the pore wall (Figure 2E and Figure 3E). From 3 M to 9 M after implantation, the mRNA expression level of COL-I showed a similar trend to that of elastin (Figure 5B). The porous SF disappeared and was completely replaced by regenerated ECM at 6 M (Figure 2F and Figure 3F). The environmental changes promoted the transition of SMCs to the contractile type, leading to a significant reduction in extracellular matrix secretion at 9 M (Figure 5A,B). In the first 6 M after implantation, the changing trend in COL-III mRNA expression level was similar to that of COL-I. In the early stage of implantation, the mRNA expression level of COL-III was lower than that of elastin and COL-I but continued to increase after 6 M (Figure 5C), having a synergistic effect on the self-assembly of COL-I fibers. The mRNA expression level of COL-III was down-regulated during the implantation period. This is consistent with the results reported by Keulen et al. [50].

### 3.4. Self-Assembly of Matrix Fibers

#### 3.4.1. SEM Observation

At 1 M after implantation, a fibrous structure had formed on both sides of the SFTS (Figure 6(A-a,A-c)) and was distributed orderly along the circumferential direction. At 3 M after implantation, fine but disordered fibers appeared and were distributed in the deposited ECM (Figure 6(B-b)) in the SFTS. At 6 M after implantation, new vascular tissue similar to the autogenous vessel was observed (Figure 6(C-a,C-b,C-c)), and the morphology of SFTS had completely disappeared. After 9 M, the matrix fibers in the regenerated vessels further self-assembled and formed a dense lamellar matrix structure (Figure 6(D-a,D-b,D-c)), indicating that the regenerated vessels were mature.

#### 3.4.2. Masson and EVG Staining Observation

Collagen fibers are the main components of the matrix. The collagen (COL-I and COL-III) fibers are closely related to the tensile strength of the vessel wall [51]. COL-I fibers with a diameter of 100–500 nm have high strength and toughness. COL-III fibers have a smaller diameter of 40–60 nm. Elastic fibers, as the name implies, are flexible fibers with a diameter of 1–10 μm. In order to visualize the characteristics of matrix fibers more clearly, Masson and WVG staining were performed in this study (Figure 7). After 1 M of implantation, fine and uneven collagen fibers (blue) appeared and were densely distributed on the outside of SFTS but lacked twists and turns (Figure 7(A-e)). Except for the center of the SFTS, collagen fibers had a dense tortuous distribution following self-assembly at 3 M (Figure 7(A-f)). With the extension of implantation time, collagen fibers in the regenerated tissue further bunched, became dense, and were distributed circumferentially in a regular curved form.

Among the ECM components, elastic fibers endow mechanical properties for the expansion and contraction of vascular tissue [52]. Compared with collagen fibers, elastic fibers were finer with high and uniformly distributed curvatures. Curved purple fibers were clearly visible by WVG staining in the intima to adventitia of autologous blood vessels, and the intima layer had a continuous elastic fiber with a convex and tortuous pattern (Figure 7B). At 1 W after the implantation of SFTS, distribution of the purple fibers was observed on the inner surface (Figure 7(B-c)). A continuous endometrial line was formed at 3 M (Figure 7(A-f)). With the prolongation of SFTSs implantation, the self-assembly and distribution of elastic fibers appeared in concert with collagen fibers. After 6 M of implantation, elastic fibers with an ordered arrangement were evenly distributed throughout the entire regenerated vascular tissue and further matured similarly to autologous blood vessels (Figure 7(A-g)).

### 3.5. Expression Level of LOXL-1 Related to Matrix Fiber Self-Assembly

LOXL-1 plays a key role in cross-linking collagen and elastin to self-assemble into matrix fibers. Here, we investigated the expression levels of LOXL-1 to confirm the maturation of collagen and elastic fibers. The positive expression of LOXL-1 was less distributed in mature autologous blood vessels and appeared more obviously on the outside of the vessel (Figure 8A). Similar to the immunohistochemical staining of collagen and elastin, non-implanted SFTSs were also non-specifically stained brown (Figure 8B), and other proteins of inflammatory cytokines and infiltration were also non-specifically stained at the initial stage of implantation (Figure 8C,D). The positive expression of LOXL-1 was low at 1 M after implantation (Figure 8E). At 3 M after implantation, the brown color was noticeably darker and more widely distributed (Figure 8F–I) and balanced in the regenerated matrix proteins (Figure 8J). The mRNA expression level of LOXL-1 at 3 M was down-regulated but significantly increased at 6 M and was up-regulated to 2.6 times that of autologous blood vessel at 9 M (Figure 8K), indicating that the matrix fibers in the regenerated vascular tissue showed positive self-assembly ability from 3 M after implantation. With the extension of implantation time, the self-assembly of matrix fibers was further enhanced to promote the maturation of regenerated tissues.

## 4. Conclusions

In this study, we report the ability and subtle repair process following implantation of a previously reported SF small-caliber, artificial blood vessel to induce tissue reconstruction in situ. In the early stage after replacement of the rabbit common carotid artery, the SFTSs provided a satisfactory microenvironment for cell growth, where SMCs actively migrated inward and exhibited high proliferative activity and matrix synthesis ability. At 1 M after implantation, new tissue was found inside and outside the SFTSs. After 3 M of implantation, the SFTS was completely filled with ECM and disordered matrix fibers. At 6 M after implantation, the new tissue completely replaced the original structure of the SFTSs and then further matured into regenerated blood vessels that were very similar to autologous blood vessels at 9 M. The SF small-caliber, artificial blood vessel reported in this study is expected to be an effective substitute in the treatment of cardiovascular diseases. The study of real-time conditions in vivo provided important guiding significance for clinical application, possible clinical emergencies, and accurate design or improvement of small-caliber artificial vessels.

## Figures and Tables

**Figure 1 polymers-14-03754-f001:**
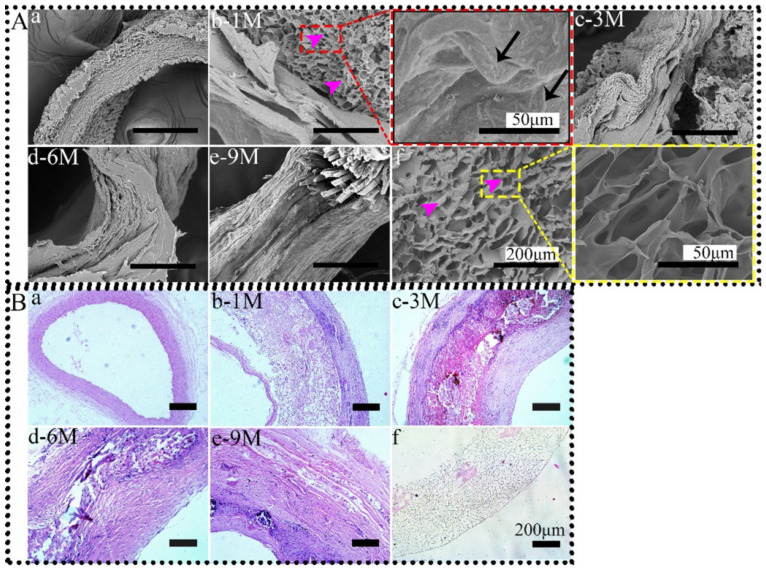
SEM images (**A**) and HE staining (**B**) of SFTSs after implantation. (**a**) Autogenous carotid artery; (**b**–**e**) represent the different time points after implantation; (**f**) non-implanted SFTS. The scale bar of magnified views is 50 μm, and others are 200 μm.

**Figure 2 polymers-14-03754-f002:**
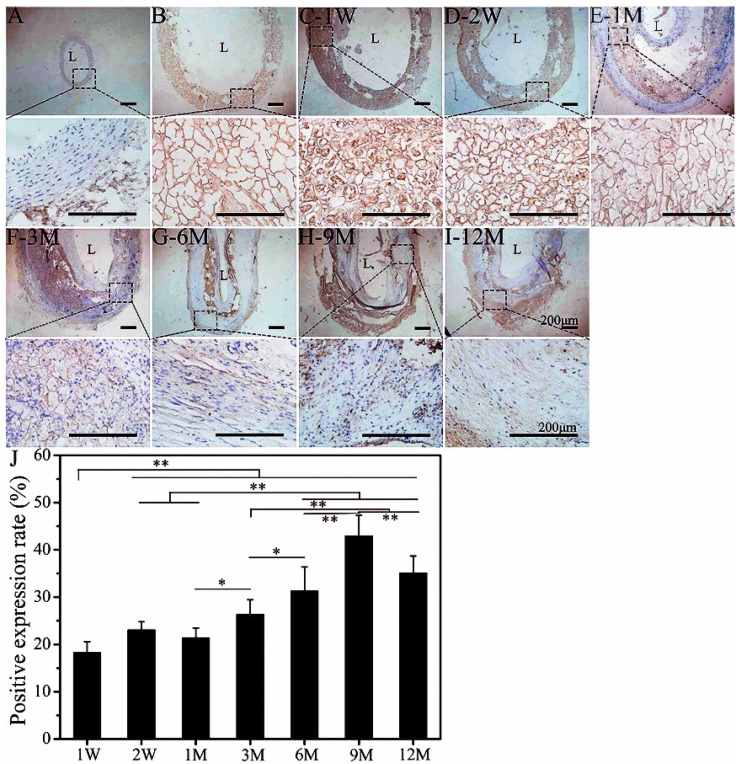
Immunoenzyme staining of elastin. (**A**): Autogenous carotid artery; (**B**): non-implanted SFTS; (**C–I**): different time points after implantation of SFTSs; (**J**): semi-quantitative results of positive expression. “L” represents the lumen. The scale bar is 200 μm. “*” represents *p* < 0.05; “**” represents *p* < 0.01.

**Figure 3 polymers-14-03754-f003:**
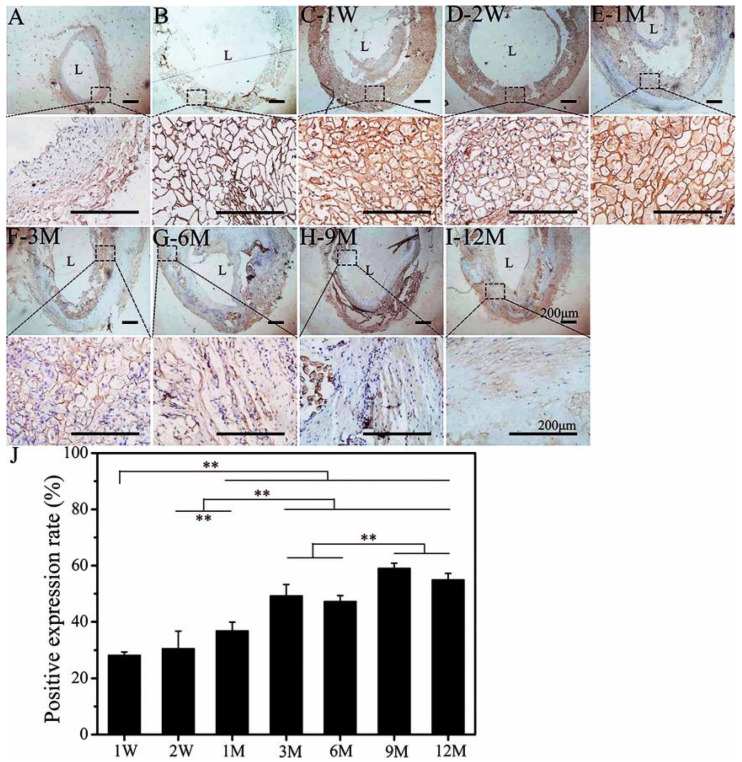
Immunoenzyme staining of COL-I. (**A**): Autogenous carotid artery; (**B**): non-implanted SFTS; (**C–I**): different time points after implantation of SFTSs; (**J**): semi-quantitative results of positive expression. “L” represents the lumen. The scale bar is 200 μm. “**” represents *p* < 0.01.

**Figure 4 polymers-14-03754-f004:**
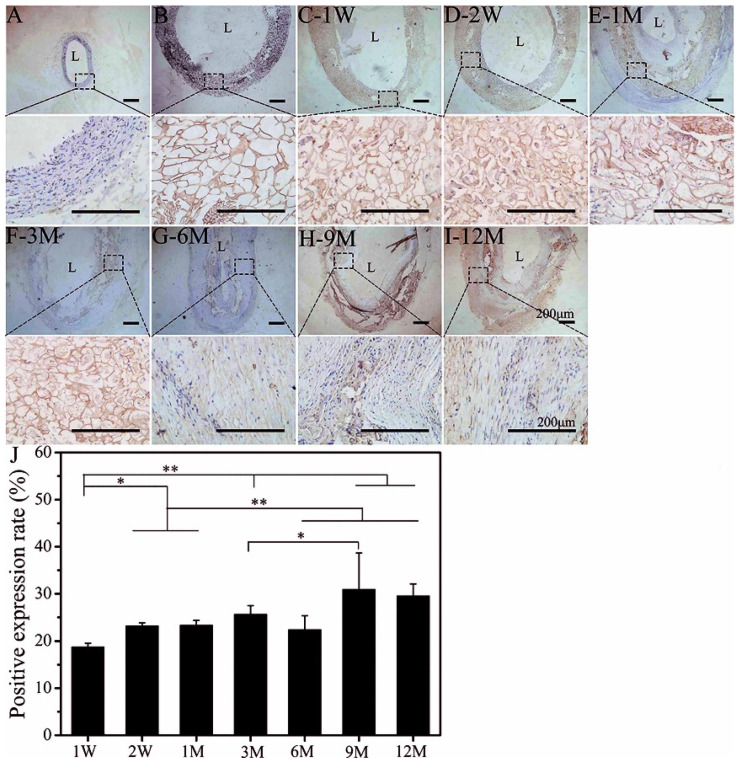
Immunoenzyme staining of COL-III. (**A**): Autogenous carotid artery; (**B**): non-implanted SFTS; (**C–I**): different time points after implantation of SFTSs; (**J**): semi-quantitative results of positive expression; “L” represents the lumen. The scale bar is 200 μm. “*” represents *p* < 0.05; “**” represents *p* < 0.01.

**Figure 5 polymers-14-03754-f005:**
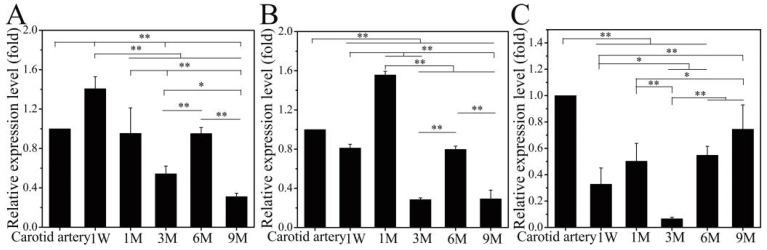
Quantitative analysis of mRNA expression level for elastin (**A**), COL-I (**B**), and COL-III (**C**). Data are presented as mean ± SD, n = 3. “*” represents *p* < 0.05; “**” represents *p* < 0.01.

**Figure 6 polymers-14-03754-f006:**
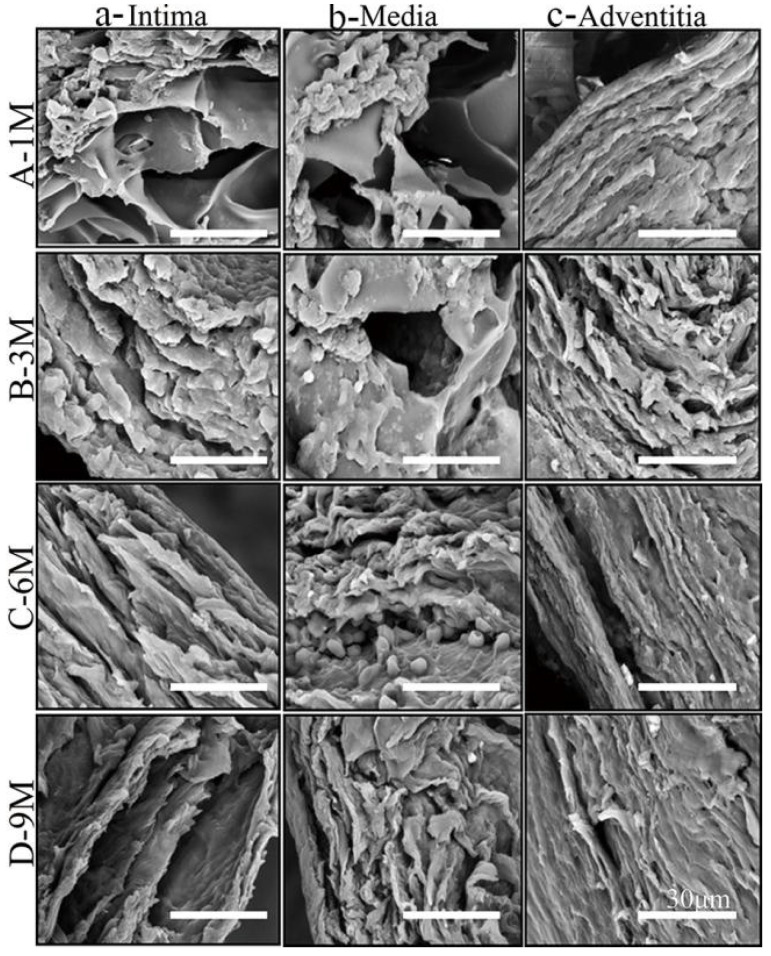
SEM images of matrix fibers in different structural layers of SFTSs after implantation. (**A**–**D**) Represent the different time points after implantation; (**a**–**c**) represent the different layers of the wall. The scale bar is 30 μm.

**Figure 7 polymers-14-03754-f007:**
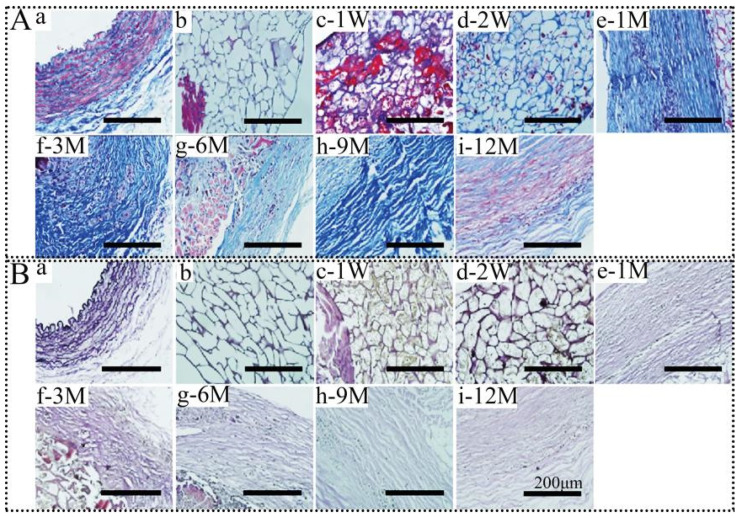
Masson (**A**) and WVG staining (**B**) of SFTSs after implantation. (**a**) Autogenous carotid artery; (**b**) non-implanted SFTS; (**c**–**i**) represent the different time points after implantation. The scale bar is 200 μm.

**Figure 8 polymers-14-03754-f008:**
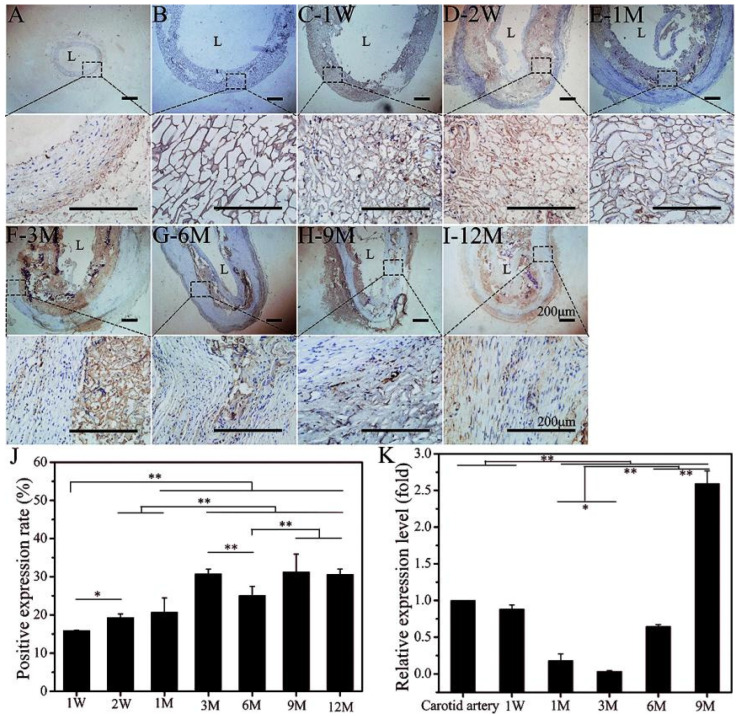
Immunoenzyme staining of LOXL-1. (**A**): Autogenous carotid artery; (**B**): non-implanted SFTS; (**C**–**I**): different time points after implantation; (**J**): semi-quantitative results of positive expression rates; (**K**): mRNA expression level of LOXL-1. Data are presented as mean ± SD, n = 3. The scale bar is 200 μm. “*” represents *p* < 0.05; “**” represents *p* < 0.01.

**Table 1 polymers-14-03754-t001:** Instructions for use of antibody.

Name	Type	Dilution Factor
Rabbit anti-human elastin	Primary antibody	400
Type III collagen antibody	Primary antibody	400
Rabbit anti-human LOXL-1 antibody	Primary antibody	400
Mouse anti-human type I collagen antibody	Primary antibody	400
Donkey anti-rabbit IgG labeled with horseradish	secondary antibody	400
Rabbit anti-mouse IgG labeled with horseradish peroxidase	secondary antibody	400

**Table 2 polymers-14-03754-t002:** Primers and their characteristics.

Gene	Primer Sequence (5′–3′)	Product Length (bps)	Annealing Temperature (°C)	Amplification Efficiency (%)
GAPDH	GTTCCACGGCAGGTCAAGG CGTACTCGGCACCAGCATCAC	119	58	94
Elastin	AAGATGGTGCAGACACTTCC AGAGCGAATCCAGCTTTGAG	116	55	99
Collagen-I	TGAGCCAGCAGATTGAGAAC CCAGTGTCCATGTCGCAGA	175	55	96
Collagen-III	AAGCCCCAGAAAATTG TGGTGGAACAGCAAAAATCA	160	52	91
LOXL-1	CCTGTGACTTCGGCAACCTCAAG GATGTCGGCGTTGTAGGTGTCG	92	53	92
CD68	CAGTTGAGAGTCGGCATTAGGTGAG CGTGAAGGATGGCAGCAGAGTG	170	55	93
CD80	AGTCGGTGAAAGAAATGGCA TAATGATGTCGGGGAAGGTG	156	56	95
CD163	CTGGGCTAATTCCAGCGCAG GATCCATCTGAGCAGGTTACTCCA	171	55	92

## Data Availability

The data presented in this study are available on request from the corresponding author.

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
