# Peer review of "Matrix Regeneration Ability In Situ Induced by a Silk Fibroin Small-Caliber Artificial Blood Vessel In Vivo"

_polymers, 2022, doi:10.3390/polym14183754_

Round 1

Reviewer 1 Report (Previous Reviewer 3)

Dear authors

There are some details listed below that the authors should address:

I hope that my comment is very useful for the improvement of the article.

(1) Please describe in detail how the SFTSs are braided. Also, although the fabricated tubes are immersed in SF aqueous solution, describe the concentration of the aqueous solution.

(2) In Figure 2 and Figure 3, the lumen of the autologous vessel image is extremely small compared to the other images. The inner diameter of the artificial blood vessel was described as 3 mm, but is there any size mismatch with the autologous blood vessel?

(3) Please describe how the authors obtained the elastin expression rate in Figure 2J.

(4) In the SEM measurement results of SFTSs, the authors describe "ECM and the gradual degradation of SFTSs, tissue regeneration is finally achieved."

I am unable to determine the degradation of SFTSs from the SEM results.

The authors should clearly indicate the degradation of SFTSs.

(5) Based on the results of RT-PCR, the authors consider that the increased elastin production in the acute phase is a phenotype in which SMCs are stimulated to proliferate and produce ECM due to contact with the artificial vessels.

This consideration is interesting, but I think there is no clear evidence for this consideration and it cannot be determined only from RT-PCR results in this study.

Do you have any data to prove that SMC is a secretory phenotype at early implantation and at 6 months?

Author Response

Reviewer 1

  • Title – Please do think better article title; please avoid descriptive title and reconsider to scientific title explaining the whole study.

Answer: We revised the title as “Matrix regeneration ability in situ induced by a silk fibroin small-caliber artificial blood vessel in vivo”.

  • Abstract – Please standardize the use term scaffold or SFTS in abstract or other sections in the manuscript.

Answer: We have standardized the use of SFTS in full text.

  • Introduction – Suggesting to include an information about current prevalence data regarding the mentioned disease/issues. “biodegradability” of SF need to be elaborate further due to the possible route or any enzyme involved in the dynamic system of human body?.

Answer: â‘  We supplemented an information about current valuation data regulating the mentioned disease/issues in the first paragraph of the introduction.

â‘¡ SF can be degraded by matrix metalloproteinase, collagenase, elastase and so on in vivo. Regenerated SF materials degrade rapidly, but high crystalline silk fibers degrade slowly. The exposition was supplemented in the third paragraph of the introduction.

  • Materials & Methods – typo error; Material & Methods-should be Materials. Please provide a figure to represent the subsection 2.2 due to unclear experimental design performed in the rabbit model. 2.5 – antibodies primary and secondary should be allocated in table together its dilution factor.

Answer: â‘  We have changed “Material & Methods” into “Materials & Methods” in the title of section 2.

â‘¡ Experimental design in the rabbit model has been described in our previous study (Steady-state behavior and endothelialization of a silk-based small-caliber scaffold in vivo transplantation. Polymers 2019, 11, 1303.), we have cited the reference (38) in original manuscript.

â‘¢ In the section 2.5, we supplemented a table (Table 1) including antibody name, type and dilution factor. 

Reviewer 2 Report (New Reviewer)

1)      Title – Please do think better article title; please avoid descriptive title and reconsider to scientific title explaining the whole study.

2)      Abstract – Please standardize the use term scaffold or SFTS in abstract or other sections in the manuscript.

3)      Introduction – Suggesting to include an information about current prevalence data regarding the mentioned disease/issues. “biodegradability” of SF need to be elaborate further due to the possible route or any enzyme involved in the dynamic system of human body?.

4)      Materials & Methods – typo error; Material & Methods- should be Materials. Please provide a figure to represent the subsection 2.2 due to unclear experimental design performed in the rabbit model. 2.5 – antibodies primary and secondary should be allocated in table together its dilution factor.

Other section looks fine.

Author Response

Reviewer 2

(1) Please describe in detail how the SFTSs are braided. Also, although the fabricated tubes are immersed in SF aqueous solution, describe the concentration of the aqueous solution.

Answer: â‘  A silk tubular fabric was braided with a 3 mm inner diameter by twisting 24 shares of 3×2 silk yarns. The 3×2 silk yarn consisted of two-ply yarns composed of two strands of thrown silk, each with three strands of 20/22 D raw silk. The exposition was supplemented in the section 2.1 of the manuscript.

â‘¡ The concentration of the SF aqueous solution is 80 mg/mL. We have supplemented it in the section 2.1 of the manuscript.

(2) In Figure 2 and Figure 3, the lumen of the autologous vessel image is extremely small compared to the other images. The inner diameter of the artificial blood vessel was described as 3 mm, but is there any size mismatch with the autologous blood vessel?

Answer: The original intention of small vessel design is to treat coronary heart disease, for example coronary artery bypass requires small vessels with an inner diameter of 3 mm. However, due to the limitations of the experimental conditions, rabbit common carotid artery (2.0~2.5 mm inner diameter) was used to carry out experiments instead of coronary artery. Therefore, the internal diameter of SFTSs is somewhat different from that of rabbit common carotid artery.

(3) Please describe how the authors obtained the elastin expression rate in Figure 2J.

Answer: Quantification was performed using Image J software (US National Institute of Health, Bethesda, Maryland, USA, http://rsb.info.nih.gov/ij/). Ten images for each time point were used to calculate the positive expression rate of matrix proteins and cytokines, in which the percentage of the brown positive expression area occupied the entire interface. The exposition has been described in the section 2.5 of original manuscript.

(4) In the SEM measurement results of SFTSs, the authors describe "ECM and the gradual degradation of SFTSs, tissue regeneration is finally achieved." I am unable to determine the degradation of SFTSs from the SEM results. The authors should clearly indicate the degradation of SFTSs.

Answer: The degradation in this study mainly refers to the degradation of the porous materials formed by regenerated silk fibroin. From the SEM image, the pore size of the porous structure became larger at 3 months compared with 1 month, which was caused by the fracture of the pore wall. At 6 months, the porous structure disappeared completely, indicating that porous SF materials were completely degraded. We supplemented pink arrows in figure 1, representing the porous structure of regenerated SF materials. This result can also be seen from figures 2F, 3F and 4F in the 3.2 section of original manuscript. At 9 months, silk fibers were still clearly visible.

(5) Based on the results of RT-PCR, the authors consider that the increased elastin production in the acute phase is a phenotype in which SMCs are stimulated to proliferate and produce ECM due to contact with the artificial vessels.

This consideration is interesting, but I think there is no clear evidence for this consideration and it cannot be determined only from RT-PCR results in this study.

Do you have any data to prove that SMC is a secretory phenotype at early implantation and at 6 months?

Answer: â‘  In the early stage of implantation, due to autologous blood vessel trauma and contact with foreign materials, smooth muscle cells transformed into secretory phenotype that proliferate and secrete extracellular matrix proteins (elastin and collagen) (Vascular smooth muscle phenotypic diversity and function. Physiol. Genomics 2010, 42A, 169–187.). This result can be seen from figure 2~4. We supplemented this reference (49) in the section 3.3 of the manuscript.

â‘¡ The porous materials began to degrade gradually after 3 months. In the following months, smooth muscle cells infiltrating into the pores proliferated and matrix proteins were deposited continuously (Figure 2~4), which also proved that smooth muscle cells were in secretory phenotype. The RT-PCR results of extracellular matrix expression corresponded with the results of immunohistochemistry.

This manuscript is a resubmission of an earlier submission. The following is a list of the peer review reports and author responses from that submission.

Round 1

Reviewer 1 Report

1.     Materials and methods are not clearly specified. How authors confirmed it is 120 D. Duration of Co-60 treatment is not specified and so forth. Concentration and amounts of reagents are also missing.

Reviewer 2 Report

The work is well done and clearly reported. The conclusions are well supported by the data and properly argued for. The research was well conducted and described, however these studies are of little concern to polymers and they are mainly focused on the use of silk fibroin in vivo (biological and tissue research is primarily described in the work). For this reason, I believe that the work should be published in another, more appropriate journal and not in "polymers".

However, the work is interesting and well written. The only comment I have is about the quality of figures (2-4 and 8) that are not very readable. 

Reviewer 3 Report

Dear authors

Thank you very much for your manuscript.

There are some details listed below that the authors should address:

I hope that my comment is very useful for the improvement of the article.

(1) In the Introduction, the authors describe that artificial polymers such as ePTFE and PET are significantly increase the adsorption of platelets and plasma proteins. 

Does silk fibroin solve these problems? Please describe the positive reasons for using silk fibroin as an artificial blood vessel.

(2) Figures 1c and d appear to have different magnification compared to the other HE images. Also, these images are unable to identify the

intima and adventitia.

 It would be better to change the images.

(3) In Figure 2E and Figure 3E, the lumen appears to be extremely blocked in the case of one month after implantation. Please make it possible to see the lumen surface.

I also thought that the implantation caused excessive thickening of the intima, could you please explain the authors' findings in this regard?

(4) In the expression of ECM-related genes by RT-PCR, COL-I and COL-III decreased over 6 months after implantation. The authors consider these to be "associated with rapid degradation of porous SF at 3M after implantation. Which result does rapid SF degradation refer to?

Also, the report states that recovery occurred at 6M, but does not discuss the reason for the significant decrease at 9M.

The authors should discuss these issues.